# Binding of Citrate-Fe^3+^ to Plastic Culture Dishes, an Artefact Useful as a Simple Technique to Screen for New Iron Chelators

**DOI:** 10.3390/ijms23126657

**Published:** 2022-06-15

**Authors:** Jiro Ogura, Toshihiro Sato, Kei Higuchi, Sathish Sivaprakasam, Jonathan Kopel, Yangzom D. Bhutia, Vadivel Ganapathy

**Affiliations:** 1Department of Cell Biology and Biochemistry, Texas Tech University Health Sciences Center, Lubbock, TX 79430, USA; jiro.ogura@med.id.yamagata-u.ac.jp (J.O.); toshihiro.sato@tohoku.ac.jp (T.S.); higuchi@toyaku.ac.jp (K.H.); sathish.sivaprakasam@ttuhsc.edu (S.S.); jonathan.kopel@ttuhsc.edu (J.K.); yangzom.d.bhutia@ttuhsc.edu (Y.D.B.); 2Graduate School of Pharmaceutical Sciences, Yamagata University, Yamagata 990-8560, Japan; 3Department of Pharmaceutical Sciences, Tohoku University Hospital, Sendai 980-8574, Japan; 4Department of Pharmaceutics, School of Pharmacy, Tokyo University of Pharmacy and Life Sciences, Tokyo 192-0392, Japan

**Keywords:** citrate-Fe^3+^ chelate, iron chelators, deferiprone, deferoxamine, 2,3-dihydroxybenzoic acid, 2,5-dihydroxybenzoic acid, carbidopa

## Abstract

NaCT mediates citrate uptake in the liver cell line HepG2. When these cells were exposed to iron (Fe^3+^), citrate uptake/binding as monitored by the association of [^14^C]-citrate with cells increased. However, there was no change in NaCT expression and function, indicating that NaCT was not responsible for this Fe^3+^-induced citrate uptake/binding. Interestingly however, the process exhibited substrate selectivity and saturability as if the process was mediated by a transporter. Notwithstanding these features, subsequent studies demonstrated that the iron-induced citrate uptake/binding did not involve citrate entry into cells; instead, the increase was due to the formation of citrate-Fe^3+^ chelate that adsorbed to the cell surface. Surprisingly, the same phenomenon was observed in culture wells without HepG2 cells, indicating the adsorption of the citrate-Fe^3+^ chelate to the plastic surface of culture wells. We used this interesting phenomenon as a simple screening technique for new iron chelators with the logic that if another iron chelator is present in the assay system, it would compete with citrate for binding to Fe^3+^ and prevent the formation and adsorption of citrate-Fe^3+^ to the culture well. This technique was validated with the known iron chelators deferiprone and deferoxamine, and with the bacterial siderophore 2,3-dihydroxybenzoic acid and the catechol carbidopa.

## 1. Introduction

We had previously identified a novel transport system for the metabolite citrate in mammalian cells that is active, Na^+^-coupled and electrogenic; it is called NaCT (Na^+^-coupled citrate transporter) [1,2,3,4]. According to the Human Genome Nomenclature, NaCT is referred to as SLC13A5 [5,6]. Recent studies have demonstrated that citrate might play a role in iron regulation in mammals; extracellular citrate induces the expression of hepcidin, the primary iron-regulatory hormone, in hepatocytes [7]. Hepcidin has a marked impact on iron homeostasis because it binds to ferroportin, the only iron exporter known in mammalian cells, and promotes its internalization and degradation [8]. When the hepcidin production in liver is compromised, circulating levels of this hormone decrease, which then leads to the increased cell-surface expression of ferroportin in the blood-facing basolateral membrane of intestinal epithelial cells and also in macrophages. The result of these changes is the increased absorption of dietary iron in the intestine and the increased release of iron from macrophages, thus increasing the circulating levels of iron and hence transferrin saturation. This consequently enhances iron uptake into cells via transferrin receptor, thus causing iron overload in multiple tissues. Hemochromatosis is a genetic disorder of iron overload that arises from mutations in the gene coding for the protein HFE, a histocompatibility antigen-like protein that is involved in iron regulation [9,10]. In this disorder, the hepatic expression of hepcidin is markedly suppressed, hence resulting in the increased absorption of dietary iron and increased release of iron from macrophages. Since extracellular citrate plays a role in the expression of hepcidin in the liver and NaCT is responsible for the entry of extracellular citrate into liver cells, we became interested in the potential regulation of NaCT by iron. Therefore, we initiated a project to examine the impact of chronic exposure to excess iron on the expression/function of NaCT and citrate uptake in the human liver cell line HepG2. These studies, however, demonstrated no evidence of the iron-dependent regulation of NaCT. However, these studies led to a surprising finding that involved the adsorption of the citrate-Fe^+^ complex to the plastic surface of the culture wells even in the absence of HepG2 cells. We used this unexpected phenomenon as the basis for a simple experimental technique to screen for new iron chelators.

## 2. Results

### 2.1. Citrate Uptake/Binding in Control and FAC (Ferric Ammonium Citrate)-Treated Liver Cells

Our original aim was to determine whether the chronic exposure of liver cells to excess iron influences the expression and function of NaCT. For this, we exposed HepG2 cells, which express NaCT [4,11,12] to ferric ammonium citrate (FAC) as an iron supplement; we cultured the cells in the presence of 65 μg/mL FAC for two passages and then used the cells for citrate uptake in the presence of NaCl to monitor NaCT function. There was a marked increase in citrate uptake/binding in HepG2 cells (Figure 1A) as a result of chronic exposure to FAC. The increase in uptake/binding was 18-fold. As FAC contains ferric ion, ammonium ion and citrate, we cultured HepG2 cells with FAC (250 μg/mL), FeCl_3_ (1 mM), NH_4_Cl (1 mM), or citrate (1 mM) for two passages, and then used the cells for citrate uptake/binding. Only treatment with FAC and FeCl_3_ increased citrate uptake/binding compared to untreated cells (Figure 1B).

### 2.2. Non-Involvement of NaCT in Citrate Uptake/Binding Induced by FAC Treatment

Human NaCT is stimulated by Li^+^ [11,12,13]. To determine whether the citrate uptake/binding that was enhanced by FAC treatment occurred via NaCT, we measured citrate uptake/binding in control and FAC-treated HepG2 cells in the absence and presence of 10 mM LiCl. In control cells, Li^+^ stimulated the citrate uptake 5-fold, as expected of the NaCT (Figure 1C). FAC treatment increased citrate uptake/binding several-fold, but the effect of Li^+^ was minimal in FAC-treated cells (Figure 1C). If the citrate uptake in control cells was subtracted from that in FAC-treated cells, the activity that was enhanced by FAC treatment was less in the presence of Li^+^. We then used the human breast cancer cell line MCF7; these cells do not express NaCT irrespective of whether or not the cells were exposed to FAC (250 μg/mL) (Figure 1D). We then used control and FAC-exposed MCF7 cells for citrate uptake/binding. In control cells, citrate uptake/binding was very low compared to HepG2 cells (0.4 ± 0.1 pmol/10^6^ cells/15 min in MCF7; 18.0 ± 0.4 pmol/10^6^ cells/15 min in HepG2) and was insensitive to Li^+^ (Figure 1E). In FAC-treated cells, citrate uptake/binding increased 6-fold and Li^+^ had little effect (Figure 2C). As there was no evidence of NaCT mRNA in MCF7 cells with or without exposure to FAC, the presence of robust citrate uptake/binding in FAC-treated cells indicates that the observed uptake/binding activity is unrelated to NaCT. In HepG2 cells that express NaCT, there was no noticeable difference in mRNA levels for the transporter between the control and FAC-treatment (Figure 1E).

### 2.3. Dose-Response Relationship for the Stimulation of Citrate Uptake/Binding by Fe^3+^

The plasma concentration of free Fe^3+^ is in a low micromolar range even under conditions of iron overload [14,15]. Therefore, to determine whether the stimulation of citrate uptake/binding could occur at concentrations of Fe^3+^ that are relevant to physiological and pathological conditions, we performed a dose-response study in MCF7 cells that do not express NaCT (Figure 2A). We found significant stimulation of citrate uptake/binding even at [Fe^3+^], as low as 5 μM when directly compared with uptake/binding in the absence of Fe^3+^ (paired Student’s *t* test; *p* < 0.05). The stimulation increased even further as [Fe^3+^] increased up to 100 μM. As iron can exist in two different ionic forms (Fe^2+^ and Fe^3+^), we tested the relative efficacy of the two forms in stimulating citrate uptake/binding in MCF7 cells. The uptake/binding of citrate to these cells increased in response to treatment with Fe^2+^ as well; however, the potency of Fe^3+^ to stimulate citrate uptake/binding was at least 2-fold greater than the potency of Fe^2+^ (data not shown). In contrast to Fe^3+^ or Fe^2+^, there was no stimulation of citrate uptake/binding by Zn^2+^ or Mn^2+^ (Figure 2B).

### 2.4. Substrate Selectivity of Fe^3+^-Stimulated Citrate Uptake/Binding

We determined the substrate selectivity of the Fe^3+^-stimulated citrate uptake/binding by monitoring the ability of various carboxylates (2.5 mM) to compete with [^14^C]-citrate (3.5 μM) for uptake/binding in the presence of FeCl_3_ (50 μM) in MCF7 cells. Only citrate and malate competed with [^14^C]-citrate for uptake/binding; lactate, pyruvate, and succinate had no effect (Figure 2C).

### 2.5. Mechanism of Fe^3+^-Stimulated Citrate Uptake/Binding

The studies described thus far suggested a novel process of Fe^3+^-stimulated citrate uptake/binding in mammalian cells that is distinct from NaCT. Even though the newly found Fe^3+^-stimulated citrate uptake/binding had key features of a carrier-mediated uptake system (e.g., substrate selectivity and substrate saturability), all our efforts to identify the transporter responsible for this uptake process failed. Then, we focused on potential alternative explanations for the phenomenon. Citrate is known for its ability to chelate Fe^3+^, and the Fe^3+^-citrate complex has a poor solubility in water [16,17,18]. In our studies, we used [^14^C]-citrate to measure uptake into cells. If citrate precipitates in the presence of Fe^3+^, the radiotracer might stick to the cells, which we could have mistaken for uptake into cells. To test this possibility, we decided to perform the same experiments using empty culture plates with no cells. Our logic was that if the citrate-Fe^3+^ chelate adsorbs to the cell surface non-specifically, it might also likely adsorb to the plastic surface of the wells in the empty culture plate. The experimental procedure was exactly the same. We added [^14^C]-citrate with or without Fe^3+^ to the empty wells, incubated for 30 min, and washed the wells with ice-cold buffer. Whatever was in the wells was then extracted with SDS/NaOH and used for measurement of radioactivity. We found markedly increased radioactivity in the extracts in the presence of Fe^3+^ than in its absence (Figure 3A). Apparently, citrate precipitates in the presence of Fe^3+^, and the precipitate sticks to the plastic wells that is not removable by washing with ice-cold buffer. However, the precipitate is easily extractable with SDS/NaOH. We then wondered how this binding of Fe^3+^-citrate precipitate would exhibit substrate saturability. To address this issue, we monitored the binding of the [^14^C]-citrate-Fe^3+^ complex to the culture wells in the presence of increasing concentrations of citrate. The binding demonstrated a hyperbolic relationship with citrate concentration, thus showing substrate saturability, as one would expect for any carrier-mediated uptake process (Figure 3B). The data perfectly conformed to the Michaelis–Menten kinetics with an apparent K_m_ value of 3.4 ± 0.3 μM (Figure 3B).

The same protocol was then used to examine the dose-response relationship for two of the well-known iron chelators, deferiprone and deferoxamine, to determine their potency to compete with citrate for chelation to Fe^3+^ in the assay system. Both compounds competed with [^14^C]-citrate for the binding to Fe^3+^ and consequently decreased the radioactivity in the extracts in a dose-dependent manner (Figure 3C). The IC_50_ values for deferiprone and deferoxamine were 20 ± 5 µM and 18 ± 4 µM, respectively. We did a similar experiment with a different class of iron chelators: the bacterial siderophore 2,3-dihydroxybenzoic acid and its structural analog 2,5-dihydroxybenzoic acid [19,20]. Both compounds competed with [^14^C]-citrate for binding to Fe^3+^ and reduced the radioactivity in the extracts, but there was a significant difference in their relative affinities for Fe^3+^ (Figure 3D). The IC_50_ values for 2,3-dihydroxybenzoic acid and 2,5-dihydroxybenzoic acid were 15 ± 2 µM and >1000 μM, respectively. 2,3-Dihydroxybenzoic acid has two hydroxyl groups on adjacent carbon atoms in the benzene ring (i.e., catechol), whereas 2,5-dihydroxybenzoic acid does not. It is most likely that this structural difference is the basis for the difference in the affinities for iron between the two compounds, because dihydroxyphenylalanine, which also contains two hydroxyl groups on adjacent carbon atoms in the benzene ring, has been demonstrated to bind iron [21,22]. If this were to be true, then the FDA-approved drug carbidopa, a compound that has a structure similar to dihydroxyphenylalanine with regard to hydroxyl groups on adjacent carbon atoms, should be able to bind iron. We tested this in the new assay method. Carbidopa was able to compete with citrate for binding to Fe^3+^; the IC_50_ value for this process was 5 ± 2 µM (Figure 3D). 

### 2.6. A Simple Screening Method for Potential Iron Chelators

Even though the present work was initiated to investigate the potential regulation of the citrate transporter NaCT by iron, the results led to unexpected findings. Citrate chelates Fe^3+^ and the resultant citrate-Fe^3+^ is a precipitate that binds to cells and empty plastic surface of the culture wells. Subsequent experiments demonstrated that the method could form the basis for a simple assay technique to screen for potential iron chelators. All the method requires is [^14^C]-citrate as a radiotracer, FeCl_3_, 24-well plastic culture dishes, and routine laboratory chemicals. The culture plates were prepared as follows. Add 2 mL of RPMI 1640 culture medium, supplemented with 10% fetal bovine serum to each well and incubate for 30 min. Then, wash the wells with 2 mL of 25 mM Hepes/Tris buffer containing 150 mM NaCl three times prior to initiation of the binding assay. Prepare 0.1 μCi [^14^C]-citrate (~4 μM) and 50 μM FeCl_3_ in 25 mM Hepes/Tris buffer containing 150 mM NaCl. Add 250 μL of this solution to each well with and without a given concentration of the candidate iron chelators. Incubate for 30 min at room temperature, followed by washing the wells with ice-cold buffer (the same buffer but without the radiotracer) twice. Then, add 500 μL of 10% SDS/0.2 N NaOH to each well and extract the contents of the well, and measure the radioactivity of the extract by scintillation spectroscopy. An iron chelator would compete with [^14^C]-citrate for binding to Fe^3+^, and decrease the amount of radioactivity associated with [^14^C]-citrate-Fe^3+^ precipitate deposited on the plastic surface of the well. Thus, any compound that decreases the radioactivity in the extract ought to be an iron chelator. If the experiment is conducted with increasing doses of the iron chelator, the affinity of the chelator for iron chelation can be determined.

## 3. Discussion

In the present study, we have discovered a simple method to screen for new iron chelators. We have used two well-characterized iron chelators, namely deferiprone and deferoxamine, to authenticate the utility of the new assay technique. We also compared the iron-chelating characteristic using this technique between the bacterial siderophore 2,3-dihydroxybenzoic acid and its structural analog 2,5-dihydroxybenzoic acid; the latter turned out to be at least 70-fold less potent as an iron chelator compared to the former. This is a surprising finding because 2,5-dihydroxybenzoic acid has been reported to the mammalian siderophore, meaning that it is the mammalian counterpart of the bacterial siderophore 2,3-dihydroxybenzoic acid, with ability to bind iron and influence iron homeostasis [19]. The relatively lower affinity of the mammalian siderophore for iron observed in the present study does not seem to support the purported biological role of this compound in mammalian cells. In this regard, it is of interest to note that BDH2, the enzyme that was demonstrated to be involved in the synthesis of the mammalian siderophore [19], is now believed to play a less likely role in the synthesis of the siderophore [23]. The findings of the present study raise doubts about the role of 2,5-dihydroxybenzoic acid as the mammalian siderophore with any significant role in iron homeostasis.

Using this new technique, we found that the carbidopa, an FDA-approved drug used in combination with laevo-DOPA for Parkinson’s disease, is a potent iron chelator. Surprisingly, the potency of carbidopa as an iron chelator is significantly higher than that of deferiprone and deferoxamine, which is used currently in clinics for iron chelation. Recent studies in our laboratory have demonstrated that carbidopa possesses significant anticancer efficacy [24]. This drug is an agonist for the nuclear receptor AhR, a characteristic that seems to contribute at least partly to its anticancer effect. The findings of the present study that carbidopa is also an iron chelator suggest that this newly found feature might also contribute to the anticancer effect. Excess iron and iron-overload diseases such as hemochromatosis seem to increase the risk for cancer [25,26,27,28]. Therefore, carbidopa might chelate iron in vivo and thus interfere with the tumor-promoting ability of excess iron, thus providing another mechanism for the drug’s anticancer efficacy. 

## 4. Materials and Methods

Materials: Ferric ammonium citrate (FAC), citrate, isocitrate, succinate, malate, lactate, pyruvate, deferiprone, and deferoxamine were purchased from Sigma-Aldrich (St. Louis, MO, USA). Radiolabeled citrate (1,5-[^14^C]-citrate; specific radioactivity, 116.4 mCi/mmol) was obtained from PerkinElmer (Waltham, MA, USA). Plastic cell culture plates were purchased from Thermo Fisher Scientific (Corning product number: 3524). RPMI culture medium 1640 was obtained from Gibco. Fetal bovine serum was from Atlanta Biologicals. 

Cell lines: HepG2 (a human hepatocellular carcinoma cell line) and MCF7 (a human estrogen receptor-positive breast cancer cell line) cells were obtained from the American Tissue Culture Collection (ATCC, Manassas, VA, USA). These cell lines were free of mycoplasma.

Uptake/binding measurements: Cells were plated in 24-well culture plates in respective culture medium appropriate for each of the cell types. When FAC-exposed cells were used, FAC was used at the same concentration when the cells were cultured for uptake/binding measurements. The medium was removed and replaced with fresh medium, again with or without FAC as appropriate, after 36 h. The next day (12 h following the medium change), the cells were used for uptake/binding measurements. The medium was aspirated and the cells were washed twice with uptake/binding medium, which was maintained at 37 °C, and then uptake/binding was initiated by the addition of radiolabeled citrate in the same medium. Incubation of the cells with radiolabeled citrate was continued for 15 min, following which the medium was removed by aspiration, and the cells washed twice with ice-cold uptake/binding medium. The cells were then solubilized with 1%SDS/0.2 N NaOH and used for measurement of radioactivity. The composition of the medium used for these measurements was: 25 mM Hepes/Tris buffer (pH 7.5) containing 140 mM NaCl, 5.4 mM KCl, 0.8 mM MgSO_4_, 1.8 mM CaCl_2_, and 5 mM glucose [1,2,3,4]. The uptake/binding measurements were made routinely in triplicates and the experiment was repeated three times. For each experiment, the concentration of protein was measured in one or two wells with cells cultured in the same 24-well plate under identical conditions as the cells used for uptake/binding, which was used for normalization among the replicates. In some experiments, the normalization was performed with the cell number instead of protein concentration.

The same technique was used to monitor the adsorption of the [^14^C]-citrate-Fe^3+^ chelate to the plastic surface of the culture wells in the absence of cells. Fresh 24-well culture plates were used. The wells were first washed with uptake/binding buffer and then [^14^C]-citrate was added to the wells with and without Fe^3+^, incubated for 30 min, and then the wells were washed with fresh uptake/binding buffer, as was conducted for the measurements with cells in the wells. Following the washing, 0.5 mL of 1%SDS/0.2 N NaOH was added to the wells and mixed. Then, the solution was pipetted out for measurement of radioactivity. 

RT-PCR: Control and FAC-exposed (250 μg/mL; 2 passages) HepG2 cells and MCF7 cells were used to prepare total RNA. RT-PCR was then used to detect the expression of NaCT (SLC13A5) mRNA. HPRT (hypoxanthine/guanine phosphoribosyltransferase) was used as an internal control. The primers used were as follows. Human NaCT: 5′-ACCTCTCTCATGCCTGTCTTGCTT-3′ (forward) and 5′-ACAACCTCTTCCGCTCTTGGTCTT-3′ (reverse); human HPRT: 5′-GCGTCGTGATTAGCGATGATGAAC-3′ (forward) and 5′-CCTCCCATCTCCTTCATGACATCT-3′ (reverse). 

Statistical analysis: In each experiment, measurements were made in triplicate and the mean of the three values was calculated for each data point. The experiments were repeated three or four times. Data (mean ± S. D.) are presented as the percent of uptake/binding in corresponding control cells. Multiple statistical comparisons were made by the one-way analysis of variance (ANOVA). Whenever appropriate, the paired Student’s *t* test was used to determine statistical significance between the control and experimental datasets. *p* < 0.05 was considered statistically significant.

## Figures and Tables

**Figure 1 ijms-23-06657-f001:**
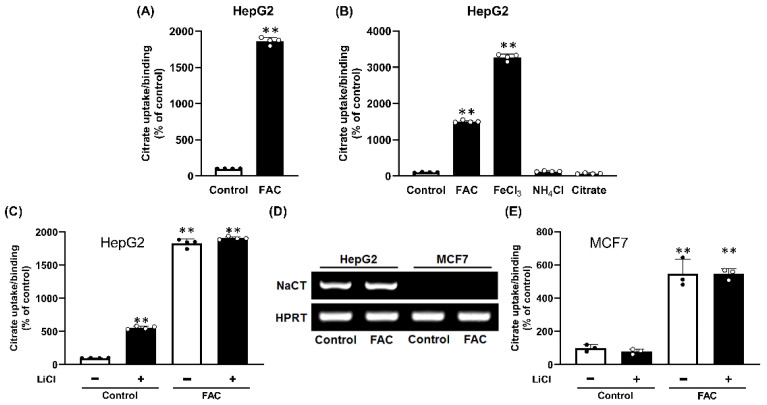
Effect of pretreatment with Fe^3+^ on citrate uptake/binding in the human hepatocarcinoma cell line HepG2 and the human breast cancer cell line MCF7. (**A**) HepG2 cells were cultured in the absence or presence of FAC (65 μg/mL) for two passages. The cells were then seeded for uptake/binding measurements and cultured in the absence or presence of FAC; confluent cells were used for [^14^C]-citrate (3.5 μM) uptake/binding (NaCl buffer, pH 7.5; 15 min incubation). (**B**) HepG2 cells were cultured in the absence or presence of FAC (250 μg/mL), FeCl_3_ (1 mM), NH_4_Cl (1 mM) or citrate (1 mM) for two passages. The cells were then seeded for uptake/binding measurements and cultured in the absence or presence of FAC, FeCl_3_, NH_4_Cl or citrate. Confluent cells were used for [^14^C]-citrate (3.5 μM) uptake/binding (NaCl buffer, pH 7.5; 15 min incubation). (**C**) To determine the potential involvement of NaCT in the Fe^3+^-stimulated citrate uptake/binding in HepG2 cells, the effect of Li^+^, an activator of NaCT transport, was studied. The cells were cultured in the absence or presence of FAC (250 μg/mL) for two passages. The cells were then seeded for uptake/binding experiments and cultured in the absence or presence of FAC; confluent cells were used for [^14^C]-citrate (3.5 μM) uptake/binding (NaCl buffer, pH 7.5; 15 min incubation) in the absence or presence of 10 mM LiCl. (**D**) Expression of NaCT mRNA in control and FAC-exposed HepG2 and MCF7 cells as assessed by RT-PCR. (**E**) Absence of Li^+^-dependent increase in citrate uptake/binding in MCF7 cells. The experimental protocol was same as the one described above for HepG2 cells. **, *p* < 0.01.

**Figure 2 ijms-23-06657-f002:**
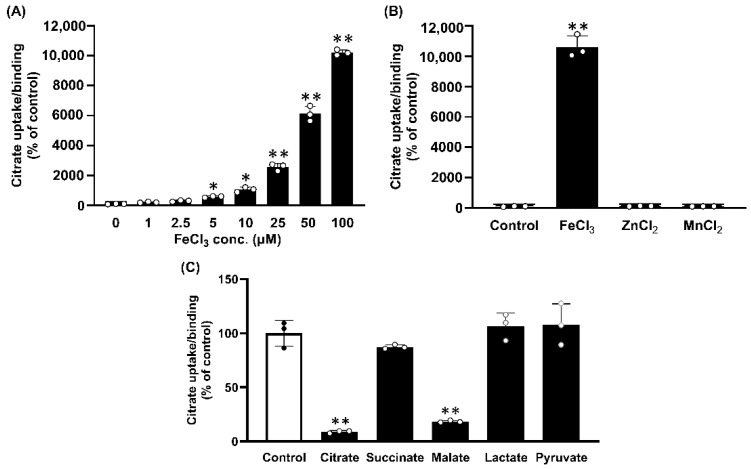
(**A**) Dose-response for the stimulation of citrate uptake/binding by Fe^3+^ in MCF7 cells. Cells were cultured to confluence and then used for measurements of [^14^C]-citrate (3.5 μM) uptake/binding (NaCl buffer, pH 7.5; 15 min incubation) in the presence of increasing concentrations of Fe^3+^ (FeCl_3_). (**B**) Effects of Zn^2+^ and Mn^2+^ on citrate uptake/binding in MCF7 cells in the presence of Na^+^. Cells were cultured to confluence and then used to measure [^14^C]-citrate (3.5 μM) uptake/binding (NaCl buffer, pH 7.5; 15 min incubation) in the absence (control) or presence of ZnCl_2_ (50 μM) or MnCl_2_ (50 μM); FeCl_3_ (50 μM) for comparison. (**C**) Substrate selectivity for Fe^3+^-stimulatable citrate uptake/binding in MCF7 cells. Cells were cultured to confluence prior to uptake/binding measurements. Uptake/binding of [^14^C]-citrate (3.5 μM) uptake/binding (NaCl buffer, pH 7.5; 15 min incubation) was measured in the presence of FeCl_3_ (50 μM) in the absence or presence of the various carboxylates (2.5 mM). *, *p* < 0.05; **, *p* < 0.01.

**Figure 3 ijms-23-06657-f003:**
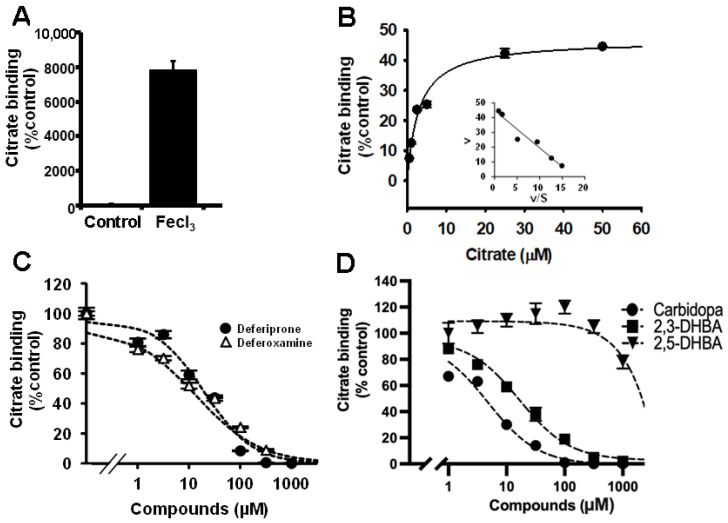
Binding of citrate-Fe^3+^ to empty culture wells in the absence of cells. (**A**) Adsorption of citrate-Fe^3+^ to the plastic surface of the wells in the culture plate was monitored with [^14^C]-citrate (3.5 μM) in NaCl buffer, pH 7.5, for 30 min in the absence and presence of FeCl_3_ (50 μM). The experimental procedure was exactly identical to the one used for uptake measurements. The wells were first exposed for 30 min to 250 μL of the NaCl buffer containing [^14^C]-citrate with and without FeCl_3_. Following this, the medium was aspirated out and the wells were washed with fresh buffer without [^14^C]-citrate and FeCl_3_, and then whatever was left in the wells was then extracted with SDS/NaOH and used for counting the radioactivity. The amount of adsorbed radioactivity in control wells without FeCl_3_ was taken as 100% and the adsorbed radiolabel in the presence of FeCl_3_ was calculated in comparison to the control value. (**B**) The wells in the culture plate were exposed for 30 min to 250 μL of the NaCl buffer containing [^14^C]-citrate (3.5 μM) in the presence of FeCl_3_ (50 μM). Unlabeled citrate was added during this exposure at increasing concentrations. The amount of citrate adsorbed to the plastic surface of the wells was then calculated as in (**A**). The same method was used to determine the dose-response relationship for the iron chelators, deferiprone and deferoxamine (**C**), and for 2,3-dihydroxybenzoic acid (2,3-DHBA), 2,5-dihydroxybenzoic acid (2,5-DHBA), and carbidopa (**D**) for their ability to inhibit the adsorption of citrate-Fe^3+^ to the plastic surface.

## Data Availability

Not applicable.

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
