# Peer review of "Binding of Citrate-Fe3+ to Plastic Culture Dishes, an Artefact Useful as a Simple Technique to Screen for New Iron Chelators"

_ijms, 2022, doi:10.3390/ijms23126657_

Round 1
Reviewer 1 Report
Major comments:
1- Uptake of citrate into HepG2 and MCF7 cells was performed by prolonged incubation with an excess of FAC (~250 µg/ml) (Figs. 1 and 2). Does this high concentration mimic physiological conditions?
2- Would it be possible to perform the dose-response relationship of FAC under the same conditions (Fig 1 and 2)?
3- Does LiCl have an effect on FeCl3-mediated citrate uptake in HepG2 cells?
4- It has been shown that FAC-induced citrate uptake is independent of NaCT in HepG2 (Fig 2). To confirm this finding, it would be interesting to examine whether inhibition or downregulation of NaCT with siRNA in the same cells shows the same trend as in Fig. 2A.
5- Is the increased citrate uptake by Fe3+ stimulation shown in Fig. 3 and Fig. 4 also independent of NaCT?
6- What is the relevance and impact of FAC versus Fe3+ in inducing the citrate uptake under physiological conditions?
7- Fe3+ is mediating citrate uptake and citrate it self is affecting Fe3+ uptake in the same cells. What would be the interplay and the regulatory mechanism of such process under physiologic and pathological conditions? A detailed discussion highlighting this issue would be very appreciated.
8- The transition to the mechanism of Fe3+ stimulated citrate uptake and the screening method for potential iron chelators (section 2.8 to section 2.9) is not smooth and clear enough. Would it be possible to improve and clarify this?
9- The authors claimed to have developed a novel method for screening potential chelators. However, the establishment process was not clearly addressed. What is the outcome of using this method in this manuscript?
Minor comments:
1- It would be desirable to minimize the number of paragraphs by combining different sections of the manuscript. For example: (Section 2.1 and 2.2 could be combined) and (Section 2.3, 2,4 and 2.5 could also be combined).
2- The title of the manuscript is not reflecting the finding and result shown in this study.
3- Line spacing formatting in section 2.1 and Discussion is inconsistent with the rest of the manuscript.
Author Response
- Initially, we performed the experiments with either 65 ug/ml or 250 ug/ml FAC. These conditions represent iron levels much greater than normal physiological levels (0.25 - 1 mM). However, the stimulation of citrate uptake/binding is observed at much lower concentrations that mimic physiological levels (Fig. 2A in the revised version).
- We did use two doses (65 ug/ml and 250 ug/ml), but we know that these levels are not physiological. That is why we repeated the uptake/binding experiments with FeCl3 using much lower concentrations, mimicking normal physiological levels of iron (Fig. 2A in the revised version).
- Li does not stimulate Fe-activated citrate uptake/binding, but it does activate citrate uptake measured in the absence of Fe but in the presence of Na (Fig. 1C in the revised version).
- We agree with the reviewer that shRNA is an appropriate approach to confirm the non-involvement of NaCT. However, we respectfully submit that such an experiment is not needed to confirm the non-involvement of NaCT because of the same Fe-induced citrate uptake/binding in MCF7 cells which do not express NaCT (Fig. 1D, E in the revised version).
- Yes.
- This point has become obsolete because of our conclusion that what we observed was not actually uptake but in fact binding of citrate-Fe to the plastic surface of the culture wells.
- The same response as above.
- We have now revised the manuscript in a major way, partly in response to the suggestions by Reviewer 2. We think that the revised version does not suffer from this drawback. We sincerely hope that the reviewer will agree after going through the revised version.
- Yes, we used this serendipitous finding of citrate-Fe binding to the plastic surface of the culture wells as the basis for developing a simple assay to monitor for new iron chelators. The outcome is described in Fig. 3 of the revised version. The method works as expected for the known iron chelators. We then used this assay to show the differential capacities of the bacterial siderophore and the so-called "mammalian" siderophore for binding to iron (Fig. 3D in the revised version).
Minor comments
- We have condensed the manuscript substantially by removing several paragraphs.
- The way the revised version is written now, we think that the title is appropriate for the manuscript. We sincerely hope that the reviewer would concur after reading the revised version.
- This has been corrected.
Reviewer 2 Report
The authors spend an extensive amount of time with assays for iron enhanced citrate "uptake", which turns out to be an artifact of binding to the plastic. They then utilize these observations to determine the efficiency of citrate as an iron chelator. That this could be an assay for identifying iron chelators is novel and should be the main emphasis of the results of the manuscript. Most of the figures presented are about the "artifact" of iron-mediated citrate binding to the plastic. I would recommend combining several of the first figures into one or two figures instead of the 8 figures provided. Figure 4 could be eliminated and comparing HepG2 versus MCF7 in figures 6 and 7 does not provide further knowledge about the assay and could be eliminated or only one shown. The manuscripts would also benefit from some editing to decrease the idea that they are measuring citrate uptake, but rather about iron enhanced citrate binding to plastic as an assay for identifying iron chelators. The experiments showing specificity of Fe3+ verses Fe2+ or Zn2+ and that citrate or malate "bind" but other carboxylates don't show that the assay works rather than that "uptake" is dependent on Fe3+.
Author Response
We have revised the manuscript substantially in response to the comments and suggestions by this reviewer. We have condensed the first 8 figures into 2 figures, retaining the data that are directly relevant to our conclusions and deleting the data that are not.
We sincerely hope that the reviewer would concur that the revised version follows exactly what he/she intended for the revision.
Reviewer 3 Report
Interesting paper - I suspect this will be of interest to the readership. Recommend no changes.
Author Response
Thank you very much for the acceptance of the original version. However, Reviewer 2 made a reasonable argument that most of the data in the original version describe the "artifact" that led to the final conclusion that what we observed was not actual uptake but in fact binding of citrate-Fe to the plastic surface of the culture wells. Therefore, he/she suggested combining the first 8 figures into 2 figures, retaining only the data that are salient to the final conclusion. Accordingly, we revised the original version in a substantial manner. We hope that this reviewer will accept this decision.
Round 2
Reviewer 2 Report
The authors have satisfied the requests by this reviewer and the manuscript is now acceptable for publication.